# 28S rRNA-Derived Fragments Represent an Independent Molecular Predictor of Short-Term Relapse in Prostate Cancer

**DOI:** 10.3390/ijms25010239

**Published:** 2023-12-23

**Authors:** Marios A. Diamantopoulos, Konstantina K. Georgoulia, Panagiotis Levis, Georgios Kotronopoulos, Konstantinos Stravodimos, Christos K. Kontos, Margaritis Avgeris, Andreas Scorilas

**Affiliations:** 1Department of Biochemistry and Molecular Biology, National and Kapodistrian University of Athens, Panepistimiopolis, 15701 Athens, Greece; imdiamantop@biol.uoa.gr (M.A.D.); georgkon@biol.uoa.gr (K.K.G.); chkontos@biol.uoa.gr (C.K.K.); mavgeris@med.uoa.gr (M.A.); 2First Department of Urology, “Laiko” General Hospital, School of Medicine, National and Kapodistrian University of Athens, 11527 Athens, Greece; levispanagiotis@gmail.com (P.L.); geoktr@hotmail.com (G.K.); kstravd@med.uoa.gr (K.S.); 3Laboratory of Clinical Biochemistry-Molecular Diagnostics, Second Department of Pediatrics, “P. & A. Kyriakou” Children’s Hospital, School of Medicine, National and Kapodistrian University of Athens, 11527 Athens, Greece

**Keywords:** prostate tumor, rRFs, cancer biomarker, clinical significance

## Abstract

Prostate cancer (PCa) is a global health concern, being a leading cause of cancer-related mortality among males. Early detection and accurate prognosis are crucial for effective management. This study delves into the diagnostic and prognostic potential of 28S rRNA-derived fragments (rRFs) in PCa. Total RNA extracted from 89 PCa and 53 benign prostate hyperplasia (BPH) tissue specimens. After 3’-end polyadenylation, we performed reverse transcription to create first-strand cDNA. Using an in-house quantitative real-time PCR (qPCR) assay, we quantified 28S rRF levels. Post-treatment biochemical relapse served as the clinical endpoint event for survival analysis, which we validated internally through bootstrap analysis. Our results revealed downregulated 28S rRF levels in PCa compared to BPH patients. Additionally, we observed a significant positive correlation between 28S rRF levels and higher Gleason scores and tumor stages. Furthermore, PCa patients with elevated 28S rRF expression had a significantly higher risk of post-treatment disease relapse independently of clinicopathological data. In conclusion, our study demonstrates, for the first time, the prognostic value of 28S rRF in prostate adenocarcinoma. Elevated 28S rRF levels independently predict short-term PCa relapse and enhance risk stratification. This establishes 28S rRF as a potential novel molecular marker for PCa prognosis.

## 1. Introduction

Prostate cancer (PCa) represents the most common form of cancer among men, ranking as the second leading cause of cancer-related deaths in Western societies [1,2]. In its early stages, PCa frequently remains asymptomatic, masking its ominous presence, while a range of distressing symptoms—spanning from obvious signs like gross hematuria to complex urinary issues and the looming threat of bone metastases—unleash their devastating effects, primarily during the disease’s advanced stages [3,4]. Understanding the origins and path of prostate tumors has been a longstanding challenge, despite recognizing crucial driver events. These include mutations deeply rooted within genes that regulate androgen production and chronic inflammatory processes that continue to exert influence [3,5,6]. Frontline therapeutic interventions deployed in the management of PCa patients predominantly revolve around the ambit of radical prostatectomy and/or the strategic implementation of androgen deprivation therapy [7]. Nonetheless, the disheartening reality persists wherein PCa, post-tumor excision, and therapeutic castration frequently ensnare individuals in a cycle of relapse, transitioning towards an androgen-independent trajectory. Thus, the dire imperative remains steadfastly anchored in the relentless pursuit of novel prognostic markers, a pressing necessity dictating the contours of personalized disease management strategies [8].

In recent scientific exploration, non-coding RNAs (ncRNAs) have emerged prominently as influential contemporary molecular markers, significantly impacting the diagnostic landscape across a spectrum of human malignancies [9,10,11]. Over the preceding decade, the ascendancy of ncRNAs has been undeniable within cellular dynamics, substantially contributing to the intricate tapestry of biological complexity and facilitating evolutionary trajectories [12]. These diverse ncRNA entities are broadly classified into two principal categories: the fundamental housekeeping cohort, encompassing essential players like rRNAs, tRNAs, snRNAs, and snoRNAs, and the regulatory counterparts—miRNAs, lncRNAs, and piRNAs—composing the intricate orchestration of cellular regulatory networks [13]. Central to the process of protein synthesis, eukaryotic ribosomes stand as foundational units housing four distinctive rRNA variants pivotal for intricate molecular assembly: 18S, 5.8S, 28S, and 5S rRNA [14]. The biogenesis of rRNA within eukaryotic cellular frameworks instigates within the nucleolus, where a triad of rRNAs (18S, 5.8S, and 28S rRNAs) undergo transcription as integral components of an extensive precursor transcript (45S), an arduous process meticulously overseen and catalyzed by the intricate machinery of RNA polymerase I (RNA Pol I [15,16]. In the context of human cellular architecture, the inception of 5S rRNA precursor (pre-5S rRNA) originates from a mosaic of genes situated in close spatial proximity to the nucleolus, orchestrated and facilitated by the transcriptional prowess of RNA polymerase III (RNA Pol III) [17,18].

In current research paradigms, rRNA-derived fragments (rRFs), notably the studied 28S rRF, stand out as a subset of small non-coding RNAs originating from the complex structure of rRNAs found in both mitochondrial and nuclear domains. Initially regarded as mere by-products resulting from rRNA metabolic processes, these mysterious non-coding RNA fragments have surpassed their initial dismissal. Instead, they have revealed an impressive ability to maintain balance across different sexes and diverse human populations [19]. Recent strides in scientific inquiry leveraging high-throughput sequencing methodologies have shed light on the complex world of small non-coding RNAs. These studies have revealed disruptions in their expression patterns, notably emphasizing the altered levels of rRFs across various human malignancies currently under investigation [20,21,22]. Moreover, antecedent explorations conducted within the confines of our esteemed research team, harnessing cutting-edge semiconductor technologies in tandem with sophisticated computational algorithms, have spearheaded innovative avenues in the identification and delineation of RNA moieties [23]. Through an exhaustive exploration encompassing fourteen distinct cellular lineages employing the prowess of small RNA-sequencing techniques, our collaborative effort uncovered a panoply of hitherto unexplored small non-coding RNA variants. Of significant note, this rigorous analysis underscored the presence of a diminutive fragment originating from the 28S rRF (archived under GenBank^®^ accession number: MT815881.1), exhibiting a substantial surge in expression levels specifically within the confines of LNCaP cells, in stark contrast to other cell lines characterized by an epithelial origin. This striking observation posits the potential pertinence of this particular fragment within the intricate landscape of PCa etiology and the cascading phenomena associated with disease progression. Recent advancements in cancer research have highlighted innovative therapeutic approaches designed to address these challenges. Notably, bioflavonoids, like hesperidin found abundantly in citrus fruits, have emerged as promising additions to cancer therapy. Hesperidin, a natural bioflavonoid, has attracted attention due to its potential to influence cell signaling pathways involved in cancer progression and treatment response [24]. Understanding how these compounds interact with cellular pathways presents new avenues for targeted therapies that may have fewer adverse effects. Additionally, ongoing research is uncovering the significance of ribosomal RNA-derived small RNAs (rsRNAs) in influencing cancer cell behavior, particularly in prostate cancer. For example, rsRNA-28S has been identified as a crucial regulator that reduces chemoresistance in prostate cancer cells by suppressing its target gene, PTGIS [25]. Gaining insights into how these rsRNAs modify gene expression could offer novel strategies to overcome treatment resistance mechanisms in cancer. The pursuit of more efficient and less harmful treatment options remains a focal point in contemporary oncology research. This study aims to evaluate the levels of 28S rRFs as potential prognostic indicators to distinguish between prostate cancer (PCa) and benign prostate hyperplasia (BPH). In addition to supplementing established clinicopathological factors, the researchers aim to clarify the prognostic significance of altered 28S rRF expression levels in predicting post-treatment disease relapse, potentially serving as an independent prognostic marker. In conclusion, this study seeks to contribute to the evolving field by investigating new molecular markers in prostate cancer, specifically focusing on fragments derived from 28S rRNA. These markers hold the potential to significantly transform prognosis methodologies in cancer care.

## 2. Results

### 2.1. Baseline Clinical Data

The age of patients in the PCa cohort ranged from 54 to 78 years, with a median age of 66 years, while in the BPH control group, it varied between 47 and 83 years, with a median age of 65 years. Among the eighty-nine PCa patients included in the study, fifty patients (56.9%) were diagnosed with pT2 tumors, thirty-eight patients (43.2%) had pT3 tumors, and for one patient, the tumor stage was unknown. Additionally, twenty-five patients (29.1%) had a Gleason score (GS) of ≤6, forty-seven patients (54.7%) had a GS of 7, and fourteen patients (16.3%) had a GS of ≥8, with GS information missing for three patients. A positive digital rectal examination (DRE) was observed in 40 PCa patients (59.7%). Lastly, seven patients (8.2%) had pre-operative serum prostate-specific antigen (PSA) levels of <4 ng/mL, fifty-five patients (64.7%) had levels between 4–10 ng/mL, and twenty-three patients (27.1%) had levels ≥10 ng/mL. Detailed clinicopathological data for PCa patients are provided in Table 1.

### 2.2. 28S rRF Expression Levels in PCa Compared to BPH Patients

In addition, we investigated the relationship between 28S rRF levels and well-established clinical prognostic markers for PCa, which include the tumor’s pathological stage, GS, and pre-operative serum PSA levels. First of all, 28S rRF expression was significantly higher in benign prostate hyperplasia (BPH) tissue specimens compared to prostate tumors (Figure 1A and Appendix A). Additionally, to assess the ability of 28S rRF expression to discriminate between benign prostate hyperplasia (BPH) and prostate tumors, we performed receiver operating characteristic (ROC) curve and logistic regression analyses (Appendix A). As illustrated by the ROC curve in Figure 1B, 28S rRF expression was able to efficiently distinguish prostate cancer from benign prostate hyperplasia (AUC = 0.688, 95% CI = 0.59–0.78, *p* < 0.001). Moreover, we found that elevated 28S rRF levels were observed in pT3 tumors in comparison to pT2 tumors (*p* = 0.017, as depicted in Figure 1C). Furthermore, although not reaching statistical significance, increased 28S rRF levels were noted in tumors with a GS of ≥8 (*p* = 0.095, as shown in Figure 1D). However, our analysis of 28S rRF levels in relation to PCa patients’ serum PSA and DRE results did not reveal a statistically significant correlation.

### 2.3. Patients with Elevated 28S rRF Levels Are at Significantly Higher Risk for Disease Short-Term Relapse Following Treatment

The median follow-up duration for PCa patients’ post-treatment was 52.0 months (95% CI: 46.98–57.02, calculated using the reverse Kaplan–Meier method). Among the 89 PCa patients initially enrolled, 14 were excluded from the survival analysis due to inadequate monitoring data. During the follow-up period, 41 patients (54.7%) experienced biochemical relapse of the disease. Kaplan–Meier survival curves demonstrated a significantly shorter disease-free survival (DFS) for PCa patients with elevated 28S rRF levels (*p* = 0.023, as illustrated in Figure 2). Furthermore, univariate Cox regression analysis confirmed an increased risk of short-term relapse for PCa patients who overexpressed 28S rRF (HR = 2.189; 95% CI: 1.089–4.397, *p* = 0.028; see Table 2).

To assess the independent prognostic value of 28S rRF for PCa relapse, multivariate Cox regression models were adjusted for factors including tumor stage, GS, patient’s PSA, DRE results, and age (detailed in Table 2). The results of the multivariate analysis (HR = 2.307; 95% CI: 1.047–5.085, *p* = 0.038) unequivocally established that elevated 28S rRF levels in the tumor could independently predict short-term relapse in PCa patients, regardless of their clinicopathological characteristics.

### 2.4. Overexpression of 28S rRF Improves Patients’ Stratification for Short-Term Relapsed Based on Established Clinical Prognostic Markers

The independent prognostic importance of elevated 28S rRF levels in predicting early PCa relapse led us to assess its potential in enhancing patient prognosis based on well-established clinical markers, such as tumor stage and GS. Kaplan–Meier survival curves vividly revealed that the combination of 28S rRF overexpression with tumor stage and GS significantly improved the stratification of patients’ risk for short-term relapse. Specifically, pT2 patients with elevated 28S rRF levels exhibited a considerably worse disease-free survival (DFS), resembling the DFS intervals of pT3 patients, in contrast to the more favorable treatment outcomes observed in pT2 patients with lower 28S rRF levels (*p* < 0.001; see Figure 3A). Similarly, elevated 28S rRF levels could distinguish GS5-7 patients with an increased risk of short-term disease relapse from GS7-9 patients with moderate 28S rRF levels (*p* < 0.001; see Figure 3B).

## 3. Discussion

PCa ranks among the most prevalent malignancies and is the second leading cause of cancer-related mortality among males worldwide [26,27]. However, PCa typically remains asymptomatic during its early stages, and it displays significant variability in patient prognosis and treatment outcomes [28,29]. Early detection of PCa is pivotal for optimizing therapeutic approaches. The identification of molecular biomarkers that can aid in early disease diagnosis and, more importantly, offer personalized prognostic information for PCa patients, is a pressing clinical priority [30,31,32]. Over the past decade, molecular markers have played a crucial role in enhancing patients’ quality of life, reducing unnecessary interventions, and containing treatment costs [33,34]. In this study, we conducted a novel investigation into the potential diagnostic and prognostic value of 28S rRF levels in predicting PCa prognosis and facilitating risk stratification. 

It is worth noting that the control of ribosome biogenesis is intricately linked to the regulation of cellular proliferation [35,36]. Additionally, rRNA transcription is influenced by various cellular pathways, with notable examples being the RAS/ERK or PI3 kinase/mTOR pathways [37,38,39,40,41,42]. In this regard, Karahan et al. have reported dysregulation in the relative expression of rRNA transcripts and the methylation of the 45S rDNA promoter in normal breast tissues when compared to breast malignancies [43]. The significance of rRNAs in the translation process is widely acknowledged [44].

The profound exploration into the intricate landscape of small regulatory RNAs has unveiled a startling revelation—the discovery of an expansive repertoire of fragments stemming from unexpected RNA reservoirs, notably including ribosomal RNAs. This paradigm shift has significantly broadened the horizons within the domain of regulatory small RNAs, fostering a newfound understanding of their intricate roles. Recent scientific revelations have, in particular, cast a luminous spotlight on fragments derived from non-coding RNAs (ncRNAs), illuminating their profound involvement in the multifaceted spectra of carcinogenesis and metastasis. Such discoveries hint at the tantalizing prospect that these molecular entities could potentially serve as invaluable candidates in the search for robust tumor biomarkers [45,46,47]. An exemplary illustration of these pivotal molecules resides within the intricate fabric of the 28S rRNA, prominently implicated in diverse cancer typologies, orchestrating pivotal roles in pivotal cellular processes like apoptosis [48]. Antecedent studies have often documented escalated rRNA levels within clinical samples harvested from diverse human malignancies, underscoring their pertinence in the oncogenic landscape [49]. However, our research diverges from this conventional narrative. Our meticulous findings portray a distinct scenario wherein the levels of 28S rRF exhibit a discernible reduction among PCa patients when juxtaposed against patients diagnosed with benign prostatic hyperplasia (BPH). This nuanced disparity could be attributed to the pivotal variance in the normalization approach adopted within our study, marked by the employment of the TATA box-binding protein (TBP) as an endogenous reference gene—a choice characterized by its documented instability in expression patterns [49]. In stark contrast, our investigation embraced a distinct normalization strategy for 28S rRF quantification, leveraging the geometric mean of two reference genes, specifically SNORD43 and SNORD48. This methodological innovation facilitated a more nuanced and precise quantification of 28S rRF expression levels, shedding light on a previously unseen vista. However, it is crucial to note certain limitations of antecedent studies. For instance, the study conducted by Uemura et al. was shackled by a notably limited sample size, encompassing merely 21 patients—a factor that understandably casts a shadow of doubt on the generalizability and broader applicability of their findings [49]. To the best of our knowledge, the diagnostic potential of 28S rRF expression in PCa has not been extensively explored. In our current research endeavor, there is compelling evidence supporting the use of 28S rRF expression levels to differentiate between prostate tumors and benign prostate tissues.

The comprehensive survival analysis conducted on a cohort of PCa patients brought to light a compelling revelation: escalated levels of 28S rRF function as a stalwart harbinger portending short-term relapse following treatment for PCa. In precise delineation, both the meticulously executed Kaplan–Meier analysis and the incisive univariate Cox analysis collectively underscored an escalated hazard, signaling an early post-resection relapse propensity among PCa patients exhibiting heightened 28S rRF levels. In further delving into the nuances of this correlation, the application of multivariate Cox regression models unearthed a significant finding: the compromised treatment outcomes observed in individuals with augmented 28S rRF levels stand as an autonomous factor, independent of a spectrum of clinicopathological variables encompassing tumor stage, GS, PSA levels, outcomes of DRE, and even patients’ chronological age. Crucially, the integration of 28S rRF levels into the evaluation framework remarkably fortified the stratification of patient risks, augmenting the predictive prowess in alignment with established prognostic clinical markers, namely, tumor stage and GS. It is pivotal to acknowledge the imperative necessity for further validation of the prognostic significance encapsulated within the expression patterns of 28S rRF. This validation must transpire within larger, independent cohorts of PCa patients, establishing a more robust and unequivocal footing for its prognostic utility. Encouragingly, our internal validation, meticulously executed through the rigorous prism of bootstrap Cox regression analysis, resiliently reaffirms the burgeoning notion: the overexpression of 28S rRF indeed emerges as a beacon of promise, a molecular beacon heralding adverse prognosis in the realm of PCa. The study’s findings hold significant implications for both clinical practice and research. Firstly, an exciting advancement in cancer biomarkers involves the discovery of 28S rRNA-derived fragments (rRFs) as potential molecular markers for diagnosing and predicting prostate cancer outcomes. This discovery opens avenues for creating more precise and tailored diagnostic tests, aiding in risk assessment and early detection for prostate cancer patients. Moreover, the link established between elevated 28S rRF levels and unfavorable post-therapy outcomes, even when accounting for traditional clinicopathological factors, underscores the potential importance of 28S rRF as an independent prognostic indicator. Such a marker could potentially enhance existing prognostic models, guiding treatment strategies and thereby enhancing patient care and management. Furthermore, integrating 28S rRF expression levels with established clinical markers demonstrates improved predictive accuracy, suggesting that a combination of markers could enhance risk assessment and treatment decisions for prostate cancer patients. Overall, the clarity surrounding the diagnostic and prognostic utility of 28S rRF in prostate cancer underscores its potential as a valuable addition to the current array of biomarkers. However, to validate its therapeutic usefulness and establish robust prognostic value, further validation in larger, independent cohorts is imperative. This step is crucial in paving the way for more effective and personalized approaches to managing prostate cancer. To underscore the significance of this study in advancing knowledge and clinical practice, it is essential to highlight the specific findings and their implications in this field.

## 4. Materials and Methods

### 4.1. Study Cohorts

Prostate tissue samples were procured from 53 individuals diagnosed with benign prostate hyperplasia (BPH) and 89 patients suffering from PCa who underwent transurethral and radical prostatectomy procedures, respectively, at the First Department of Urology, National and Kapodistrian University of Athens, “Laiko General Hospital”, located in Athens, Greece. Our study excluded patients who had undergone hormonal therapy or radiotherapy prior to surgery. For the post-treatment survival analysis, we successfully tracked 75 PCa patients, while 14 individuals were excluded from the survival analysis due to unclear monitoring data, the administration of adjuvant therapy before disease recurrence, and/or the presence of positive surgical margins. Biochemical relapse, defined as two consecutive serum PSA measurements of ≥0.2 ng/mL, served as the clinical endpoint event.

The study received institutional approval from the ethical committee at “Laiko General Hospital” and adhered to the ethical principles outlined in the 1995 Declaration of Helsinki, as revised in 2008. Informed consent was obtained from all participating patients.

### 4.2. Cancer Cell Line Culture 

LNCaP cells, derived from human prostate carcinoma, were cultivated in RPMI-1640 medium (ThermoFisher Scientific, Massachusetts, MA, USA) supplemented with 10% fetal bovine serum, 0.1 g/L streptomycin, 100 kU/L penicillin, and 2 mM L-glutamine. The cells were initially plated at a density of 5 × 10^4^ cells/mL and incubated in a humidified environment with 5% CO_2_ at 37 °C for 48 h before harvesting.

### 4.3. Total RNA Extraction 

Prostate tissue specimens and LNCaP cells were homogenized and subjected to RNA extraction using TRI Reagent (Molecular Research Center Inc., Cincinnati, OH, USA) following the manufacturer’s guidelines. The purity and concentration of the extracted RNA were assessed at 260 and 280 nm using the BioSpec-nano Micro-volume UV-Vis Spectrophotometer (Shimadzu Corp., Kyoto, Japan). Additionally, RNA integrity was evaluated through agarose gel electrophoresis [50].

### 4.4. Total RNA Polyadenylation and First-Strand cDNA Synthesis

Polyadenylation of the RNA 3′-end was carried out in a 10 μL reaction using 1U of recombinant E. coli poly(A) polymerase (New England Biolabs Ltd., Ontario, ON, Canada), 800 μM ATP, and 1 μg of total RNA template. The reaction was conducted at 37 °C for 60 min, followed by enzyme inactivation at 65 °C for 10 min. The polyadenylated RNA then served as a template for the first-strand cDNA synthesis in a 20 μL reaction, which included 50U of M-MLV reverse transcriptase (Invitrogen, Thermo Fisher Scientific Inc., Waltham, MA, USA), 40U of RNaseOUT recombinant ribonuclease inhibitor (Invitrogen), and 0.25 μM of the poly(T)-adapter 5′-GCGAGCACAGAATTAATACGACTCACTATAGGTTTTTTTTTTTTVN-3′ (N = G, A, T, C and V = G, A, C). This reaction was performed at 37 °C for 60 min, and the MMLV enzyme was subsequently inactivated at 70 °C for 15 min [51].

### 4.5. Quantitative Real-Time PCR (qPCR)

To quantify 28S rRF expression, a quantitative real-time PCR (qPCR) assay using SYBR Green as the detection method was developed. Specific forward primers were designed for both 28S rRF (with GenBank^®^ accession number: MT815881.1) and small nucleolar RNA, C/D box 48 (*SNORD48*, also known as *RNU48*, with GenBank^®^ accession number: NR_002745.1). These forward primers were paired with a universal reverse primer. The use of the 28S rRF-specific forward primer (5′-TCTGGGTCGGGGTTTCGTA-3′) and the *SNORD48*-specific forward primer (5′-TGATGATGACCCCAGGTAACTCT-3′) along with the universal reverse primer (5′-GCGAGCACAGAATTAATACGAC-3′, complementary to the oligo-dT-adapter) generated amplicons of 67 bp for 28S rRF and 105 bp for *SNORD48*.

The qPCR assays were conducted using the 7500 Real-Time PCR system (Applied Biosystems, Foster City, CA, USA). Each reaction included 5 ng of cDNA, 5 μL of KAPA SYBR FAST qPCR master mix from Kapa Biosystems Inc. (Woburn, MA, USA), and 300 nM of each primer, resulting in a final reaction volume of 10 μL. The thermal protocol comprised an initial denaturation step at 95 °C for 3 min, followed by 40 cycles of denaturation at 95 °C for 3 s and primer annealing/extension at 60 °C for 30 s. All samples were analyzed in duplicate for each target, and the relative quantification of 28S rRF was carried out using the 2-ΔCT method. This method used *SNORD48* as the endogenous reference gene for normalization and the LNCaP cell line cDNA as the calibrator [52].

### 4.6. Statistical Analysis

We assessed the normal distribution of 28S rRF levels in PCa and BPH cohorts using the Kolmogorov–Smirnov and Shapiro–Wilk tests. Since the data did not follow a normal distribution, we compared 28S rRF differences between PCa and BPH cohorts, as well as among PCa categorical variables, using non-parametric tests, specifically the Mann-Whitney *U* and Kruskal-Wallis tests. To determine the discriminatory significance of 28S rRF in PCa, we conducted a receptor operating characteristic (ROC) curve analysis, following the Hanley and McNeil method, and performed logistic regression analysis. For the survival analysis of PCa patients, we utilized Kaplan–Meier curves and assessed them with the log-rank test. We also applied Cox proportional regression models. Internal validation was carried out using bootstrap analysis, which involved generating 1000 bootstrap samples. To establish the optimal cut-off value for 28S rRF levels and categorize PCa patients into high- and low-28S rRF groups, we employed the X-tile program [53,54].

## 5. Conclusions

PCa poses a significant health challenge globally, demanding effective diagnostic and prognostic strategies for improved patient management. This study unveiled the diagnostic and prognostic potential of 28S rRNA-derived fragments (rRFs) in PCa, shedding light on their role as molecular markers for disease stratification and prognosis. A distinct downregulation of 28S rRF levels in PCa compared to benign prostate hyperplasia (BPH) patients was observed, with a positive correlation noted between elevated 28S rRF expression and higher Gleason scores and tumor stages. Notably, patients exhibiting increased 28S rRF levels faced a significantly elevated risk of short-term disease relapse post-treatment, independent of conventional clinicopathological variables. Integration of 28S rRF expression augmented risk stratification, enhancing the predictive power alongside established clinical markers, such as tumor stage and Gleason score. The mentioned findings underscore the potential of 28S rRF as a novel molecular marker for PCa prognosis. However, further validation within larger independent cohorts is imperative to solidify its clinical utility and establish robust prognostic value. Despite the limitations and complexities inherent in PCa prognosis, the identification of 28S rRF as a potential biomarker signifies a promising avenue for refining personalized treatment strategies and advancing the clinical management of prostate cancer patients.

In conclusion, this study sheds light on the prognostic potential of rRFs in PCa, highlighting their role as molecular markers for disease stratification and prognosis. The observed decrease in 28S rRF levels in PCa compared to benign prostate hyperplasia (BPH) patients, coupled with their correlation with clinicopathological factors and short-term relapse, underscores their promising utility in clinical settings. Future research could explore several areas to strengthen the clinical relevance of 28S rRFs. Firstly, rigorous validation in larger, diverse cohorts of PCa patients across multiple centers is crucial to ensure robustness, applicability across populations, and broader clinical use. Long-term prospective studies would be invaluable in examining how 28S rRF levels change over the disease course, offering insights into their usefulness for long-term prognostication and monitoring treatment responses. Moreover, delving into the mechanisms that regulate the dysregulation of 28S rRF in PCa could provide deeper molecular insights. Understanding how these fragments influence key pathways involved in tumor progression might reveal new therapeutic targets. Additionally, establishing standardized assays and integrating 28S rRF with other validated biomarkers in a comprehensive panel could refine risk assessment and guide personalized treatment strategies. Lastly, combining 28S rRF analysis with other omics data—such as genomics, transcriptomics, and proteomics—may uncover complex networks and signatures that better predict disease aggressiveness and treatment response. This multidimensional approach could offer a more comprehensive understanding of PCa behavior.

## Figures and Tables

**Figure 1 ijms-25-00239-f001:**
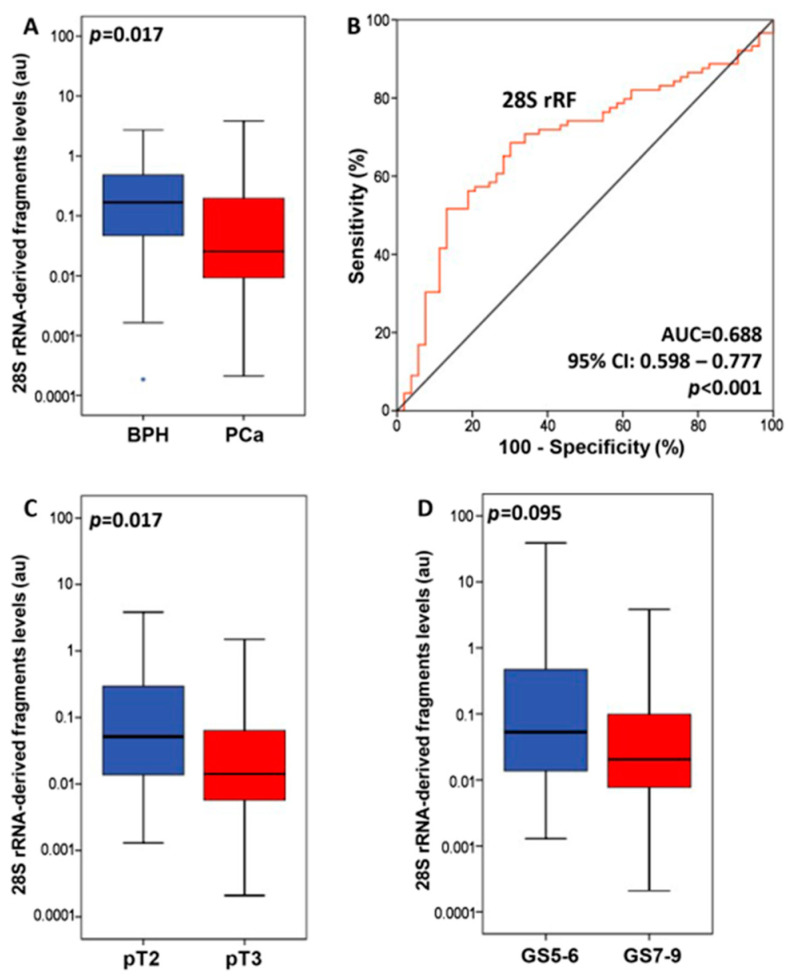
Comparison of patients’ subgroups, based on the 28S rRF levels. (**A**) Comparison of 28S rRF levels in prostate tumors (PCa) and benign prostate hyperplasia (BPH) tissue specimens. The asterisk (*) symbol indicates an outlier of the distribution. (**B**) Receiver operating characteristic (ROC) curve analysis for 28S rRF levels in discriminating PCa from BPH patients. (**C**) Comparison of 28S rRF levels with prostate tumors’ pathological stage. (**D**) Comparison of 28S rRF levels with prostate tumors’ Gleason score. *p* values calculated by Mann–Whitney *U* test (**A**,**C**,**D**) or Hanley and McNeil method (**B**).

**Figure 2 ijms-25-00239-f002:**
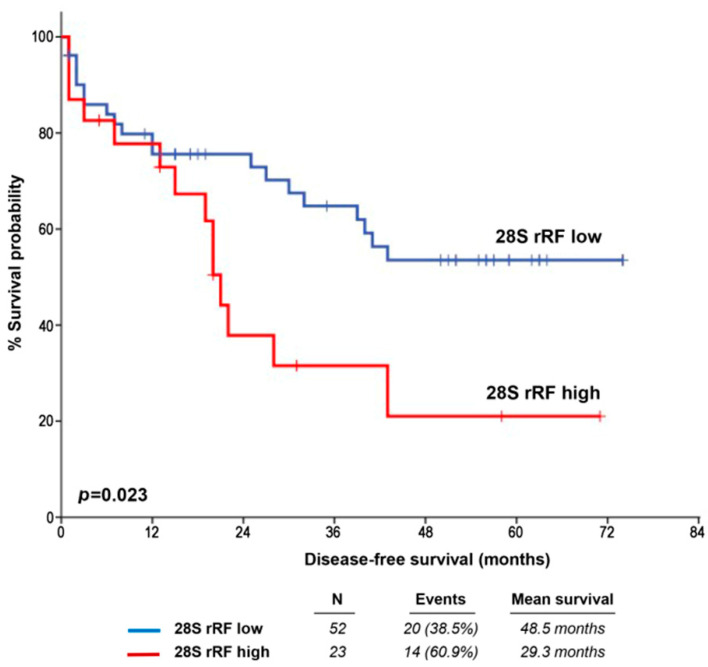
Kaplan–Meier survival curves for the disease-free survival (DFS) of PCa patients according to 28S rRF levels. *p* value calculated by log-rank test.

**Figure 3 ijms-25-00239-f003:**
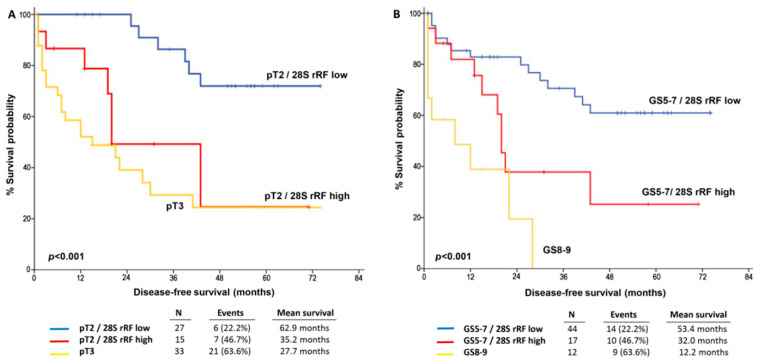
Kaplan–Meier survival curves for the disease-free survival (DFS) of PCa patients according to 28S rRF levels complied with tumor stage (**A**) and Gleason score (**B**). *p* values calculated by log-rank test.

**Table 1 ijms-25-00239-t001:** Clinicopathological data of PCa patients.

Variable	Number of PCa Patients *(n* = 89)
Gleason score ^1^	
≤6	25 (29.1%)
7	47 (54.6%)
≥8	14 (16.3%)
Missing data	3
Tumor stage ^2^	
pT2a	18 (20.4%)
pT2b	16 (18.2%)
pT2c	16 (18.2%)
pT3a	22 (25.0%)
pT3b	16 (18.2%)
Missing data	1
DRE	
Positive	40 (59.7%)
Negative	27 (40.3%)
Missing data	22
Serum PSA levels	
<4 ng/mL	7 (8.2%)
4–10 ng/mL	55 (64.7%)
≥10 ng/mL	23 (27.1%)
Missing data	4
Age	
<65 years	38 (43.7%)
65–74 years	43 (49.4%)
≥75 years	6 (6.9%)
Missing data	2

^1^ Gleason score ≤ 6: low-grade cancer; a score of 7: medium-grade cancer; and a score ≥ 8: high-grade cancer. ^2^ Tumor stage (pT2a: tumor invasion in one half (or less) of one side of prostate; pT2b: tumor involving more than one half of one lobe; pT2c: tumor involving both lobes of prostate gland; pT3a: tumor has broken through capsule; pT3b: tumor extends to pelvic wall and/or causes hydronephrosis or non-functional kidney).

**Table 2 ijms-25-00239-t002:** Cox regression analysis for the prediction of PCa patients’ disease-free survival (DFS) according to 28S rRF levels.

Covariate	Univariate Analysis					Multivariate Analysis ^6^				
	HR ^1^	95% CI ^2^	*p* Value ^3^	BCa ^4^ Bootstrap ^5^ 95% CI ^2^	Bootstrap ^5^ *p* Value ^3^	HR ^1^	95% CI ^2^	*p* Value ^3^	BCa ^4^ Bootstrap ^5^ 95% CI ^2^	Bootstrap ^5^ *p* Value ^3^
28S rRF levels										
High	2.189	1.089–4.397	*0.028*	1.113–4.358	*0.016*	2.307	1.047–5.085	0.038	0.880–7.221	*0.042*
Gleason score										
≥7	2.135	0.924–4.938	0.076	1.011–6.297	*0.048*	1.219	0.459–3.241	0.691	0.059–9.485 × 10^4^	0.716
Tumor stage										
≥T3a	3.549	1.759–7.163	*<0.001*	1.794–9.328	*0.001*	2.736	1.087–6.890	0.033	0.612–41.18	0.075
PSA	1.091	1.027–1.160	*0.005*	1.009–1.262	*0.014*	0.998	0.911–1.092	0.958	0.884–1.164	0.971
Age	0.997	0.946–1.050	0.901	0.946–1.049	0.888	0.580	0.927–1.044	0.983	0.912–1.050	0.624
DRE										
Positive	2.276	1.061–4.883	*0.035*	1.157–5.403	*0.017*	2.162	0.942–4.962	0.069	0.798–7.830	0.083

^1^ Hazard ratio. ^2^ Confidence interval. ^3^ Italics indicate a significant *p* value. ^4^ Bias-corrected and accelerated. ^5^ Based on 1000 bootstrap samples. ^6^ Multivariate analysis was adjusted for patients’ 28S rRF levels, Gleason score, tumor stage, serum PSA levels, age, and DRE.

## Data Availability

The data presented in this study are available on request from the corresponding author. The data are not publicly available due to ethical issues.

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
