# Peer review of "28S rRNA-Derived Fragments Represent an Independent Molecular Predictor of Short-Term Relapse in Prostate Cancer"

_ijms, 2023, doi:10.3390/ijms25010239_

Round 1

Reviewer 1 Report

Comments and Suggestions for Authors

This study presents an interesting finding to include 28S rRNA-derived fragments as the novel  prostate cancer biomarkers. The study is reasonably designed and the manuscript is well-written. I have only two minor concerns:

1) The major topic is about the prognosis of  prostate cancer, therefore, I strongly recommend the authors to remove the 'diagnosis' term in the last sentence of the abstract.

2) HR=1.000 in Table 2 is not necessary but a bit confusing. This can be removed.

Author Response

ReVIEWER’S Comments and Corresponding Responses

Reviewer #1 (Comments to the Author):

  1. The major topic is about the prognosis of prostate cancer; therefore, I strongly recommend the authors to remove the 'diagnosis' term in the last sentence of the abstract.

The authors wish to thank the Reviewer for their constructive comments that led to the improvement of the current manuscript. We deeply appreciate the Reviewer’s suggestion to remove the 'diagnosis' term in the last sentence of the abstract. Therefore, the authors removed the 'diagnosis' term in the last sentence of the abstract. We sincerely hope that now the topic is clearer for the reader.

  1. HR=1.000 in Table 2 is not necessary but a bit confusing. This can be removed.

We deeply appreciate the Reviewer’s suggestion to remove HR=1.000 in Table 2. Therefore, the authors removed HR=1.000 in Table 2. We sincerely hope that Table 2 is now clearer for the reader.

The authors wish to thank the Reviewers for their constructive comments that led to the improvement of the current manuscript.

Reviewer 2 Report

Comments and Suggestions for Authors

The presented manuscript is about prostate cancer (PCa),as a global health concern, being a leading cause of cancer-related mortality among males.This study delves into the diagnostic and prognostic potential of 28S rRNA-derived fragments (rRFs) in PCa. Using an in-house quantitative real-time PCR (qPCR) assay, the authors quantified 28S rRF levels. Post-treatment biochemical relapse served as the clinical endpoint event for survival analysis, which they validated internally through bootstrap analysis. Their results revealed downregulated 28S rRF levels in PCa compared to BPH patients. It is generally well writen and the content of the pubblication is compared to other scientific pubblications.

However, there are some sencentences, which need modifications previous to acceptance for pubblications.

Author Response

Reviewer #2 (Comments to the Author):

  1. Please revise this sentence(s), because the acronym PCa has been explained once in one of the sentences of the paragraph (peer-review-33799932.v2.pdf)

I am glad to inform you that the authors modified the mentioned sentences that you mentioned in .pdf file as well as the abbreviations in the whole manuscript.

The authors wish to thank the Reviewers for their constructive comments that led to the improvement of the current manuscript.

Reviewer 3 Report

Comments and Suggestions for Authors

I found this as a solid work and its quality is good. However, it will be great, if the authors are able to address some minor issues.

1.      Introduction may be improved adding new information in order to provide an adequate state-of-the-art.

i.                    Cancer represents one of the most frequent causes of death in the world. The current therapeutic options, including radiation therapy and chemotherapy, have various adverse effects on patients’ health.

Hesperidin, a Bioflavonoid in Cancer Therapy: A Review for a Mechanism of Action through the Modulation of Cell Signaling Pathways. https://doi.org/10.3390/molecules28135152.

ii.                  rsRNA-28S attenuates prostate cancer cell chemoresistance by downregulating its target gene PTGIS.

rRNA-Derived Small RNA rsRNA-28S Regulates the Chemoresistance of Prostate Cancer Cells by Targeting PTGIS.

https://doi.org/10.31083/j.fbl2805102.

2.      Conclusion should be brief and may include future research studies secondary to the current findings of this study.

3.      Why does ctDNA provide a much more holistic view of tumor characteristics and progression emanating from primary and metastasized tumor foci?

4.      Provide the significance of study at the end of discussion.

5.      Ligand of table 1 should be more informative.

6.      In the materials and methods section, method lack references.

7.      The objectives should be better explained at the end of introduction.

8.      How many primers are needed for the process of reverse transcription of RNA?

9.      What is target length of M-MLV reverse transcriptase?

10.  Why is SYBR green I assay applied to a number of conditions where a tumour suppressor gene is involved in inherited susceptibility to cancer?

11.  What are the relations between threshold cycles and relative gene expression?

Comments on the Quality of English Language

English spelling should be double-checked.

Author Response

Reviewer #3 (Comments to the Author):

  1. Introduction may be improved adding new information in order to provide an adequate state-of-the-art. i. Cancer represents one of the most frequent causes of death in the world. The current therapeutic options, including radiation therapy and chemotherapy, have various adverse effects on patients’ health. Hesperidin, a Bioflavonoid in Cancer Therapy: A Review for a Mechanism of Action through the Modulation of Cell Signaling Pathways. https://doi.org/10.3390/molecules28135152. ii. rsRNA-28S attenuates prostate cancer cell chemoresistance by downregulating its target gene PTGIS. rRNA-Derived Small RNA rsRNA-28S Regulates the Chemoresistance of Prostate Cancer Cells by Targeting PTGIS. https://doi.org/10.31083/j.fbl2805102.

We would like to thank you for your constructive comments that led to the improvement of the current manuscript. The following paragraph (Lines 95-117) was added to provide an adequate state-of-the-art.: “Recent advancements in cancer research have highlighted innovative therapeutic approaches designed to address these challenges. Notably, bioflavonoids, like hesperidin found abundantly in citrus fruits, have emerged as promising additions to cancer therapy. Hesperidin, a natural bioflavonoid, has attracted attention due to its potential to influence cell signaling pathways involved in cancer progression and treatment response (Hesperidin, a Bioflavonoid in Cancer Therapy: A Review for a Mechanism of Action through the Modulation of Cell Signaling Pathways). Understanding how these compounds interact with cellular pathways presents new avenues for targeted therapies that may have fewer adverse effects. Additionally, ongoing research is uncovering the significance of ribosomal RNA-derived small RNAs (rsRNAs) in influencing cancer cell behavior, particularly in prostate cancer. For example, rsRNA-28S has been identified as a crucial regulator that reduces chemoresistance in prostate cancer cells by suppressing its target gene, PTGIS (rRNA-Derived Small RNA rsRNA-28S Regulates the Chemoresistance of Prostate Cancer Cells by Targeting PTGIS). Gaining insights into how these rsRNAs modify gene expression could offer novel strategies to overcome treatment resistance mechanisms in cancer. The pursuit of more efficient and less harmful treatment options remains a focal point in contemporary oncology research. This study aims to evaluate the levels of 28S rRFs as potential prog-nostic indicators to distinguish between prostate cancer (PCa) and benign prostate hyperplasia (BPH). In addition to supplementing established clinicopathological factors, the researchers aim to clarify the prognostic significance of altered 28S rRF expression levels in predicting post-treatment disease relapse, potentially serving as an independent prognostic marker. In conclusion, this study seeks to contribute to the evolving field by investigating new molecular markers in prostate cancer, specifically focusing on fragments derived from 28S rRNA. These markers hold the potential to significantly transform prognosis methodologies in cancer care

  1. The conclusion should be brief and may include future research studies secondary to the current findings of this study.

We would like to thank you for your constructive comments that led to the improvement of the current manuscript. The following paragraph (Lines 393-414) was added to include future research studies: In conclusion, this study sheds light on the prognostic potential of rRFs in PCa, highlighting their role as molecular markers for disease stratification and prognosis. The observed decrease in 28S rRF levels in PCa compared to benign prostate hyperplasia (BPH) patients, coupled with their correlation with clinicopathological factors and short-term relapse, underscores their promising utility in clinical settings.Looking ahead, future research could explore several areas to strengthen the clinical relevance of 28S rRFs. Firstly, rigorous validation in larger, diverse cohorts of PCa patients across multiple centers is crucial to ensure robustness, applicability across populations, and broader clinical use. Long-term prospective studies would be invaluable in examining how 28S rRF levels change over the disease course, offering insights into their usefulness for long-term prognostication and monitoring treatment responses. Moreover, delving into the mechanisms that regulate the dysregulation of 28S rRF in PCa could provide deeper molecular insights. Understanding how these fragments influence key pathways involved in tumor progression might reveal new therapeutic targets. Additionally, establishing standardized assays and integrating 28S rRF with other validated biomarkers in a comprehensive panel could refine risk assessment and guide personalized treatment strategies. Lastly, combining 28S rRF analysis with other omics data—such as genomics, transcriptomics, and proteomics—may uncover complex networks and signatures that better predict disease aggressiveness and treatment response. This multi-dimensional approach could offer a more comprehensive understanding of PCa behavior.

  1. Why does ctDNA provide a much more holistic view of tumor characteristics and progression emanating from primary and metastasized tumor foci?

We would like to thank you for your constructive comments that led to the improvement of the current manuscript. Compared to traditional biopsy-based analysis, cell-free tumor DNA (ctDNA) provides a more thorough assessment of tumor features and progression for a number of reasons. DNA fragments released into the bloodstream by tumor cells are represented by ctDNA. The genetic material from the primary and metastatic tumor locations is carried by this DNA. By reflecting the genetic changes that occur at various locations and throughout time, ctDNA offers a more comprehensive picture of the heterogeneity and evolution of tumors as they grow and spread. Because of spatial heterogeneity, obtaining tissue biopsies from primary and metastatic sites can be difficult, invasive, and may not fully capture the tumor landscape.Contrarily, ctDNA is readily obtained with a simple blood draw, enabling repeated, non-invasive sampling throughout time. This makes it easier to track tumor changes more dynamically. Tumors, both primary and metastatic, can show notable genetic heterogeneity. The genetic diversity between several tumor lesions may not be accurately represented by biopsies taken from a single site. Because ctDNA includes a mixture of DNA fragments from different tumor sites, it provides a more thorough picture of this heterogeneity. It is possible to detect therapy response, disease progression, or recurrence in real time by keeping an eye on ctDNA levels and mutations. Therapeutic modifications can be made promptly by considering changes in ctDNA levels or specific mutations that may signal treatment success or the emergence of resistance. Compared to conventional imaging or biomarkers, ctDNA detection may be able to identify the existence or recurrence of a tumor early. To enable a more accurate evaluation of the disease status, it can also be utilized to monitor minimum residual disease following surgery or treatment. All things considered, ctDNA analysis offers a more comprehensive, real-time, and non-invasive way to comprehend tumor properties, heterogeneity, and treatment response across several tumor sites, giving a more comprehensive picture of the disease's progression and evolution.

  1. Provide the significance of study at the end of discussion.

We would like to thank you for your constructive comments that led to the improvement of the current manuscript. The following paragraph (Lines 273-293) was added to provide the significance of study at the end of discussion: The study's findings hold significant implications for both clinical practice and research. Firstly, an exciting advancement in cancer biomarkers involves the discovery of 28S rRNA-derived fragments (rRFs) as potential molecular markers for diagnosing and predicting prostate cancer outcomes. This discovery opens avenues for creating more precise and tailored diagnostic tests, aiding in risk assessment and early detection for prostate cancer patients. Moreover, the link established between elevated 28S rRF levels and unfavorable post-therapy outcomes, even when accounting for traditional clinicopathological factors, underscores the potential importance of 28S rRF as an independent prognostic indicator. Such a marker could potentially enhance existing prognostic models, guiding treatment strategies and thereby enhancing patient care and management. Furthermore, integrating 28S rRF expression levels with established clinical markers demonstrates improved predictive accuracy, suggesting that a combination of markers could enhance risk assessment and treatment decisions for prostate cancer patients. Overall, the clarity surrounding the diagnostic and prognostic utility of 28S rRF in prostate cancer underscores its potential as a valuable addition to the current array of biomarkers. However, to validate its therapeutic usefulness and establish robust prognostic value, further validation in larger, independent cohorts is imperative. This step is crucial in paving the way for more effective and personalized approaches to managing prostate cancer. To underscore the significance of this study in advancing knowledge and clinical practice, it is essential to highlight the specific findings and their implications in this field.

  1. Ligand of table 1 should be more informative.

We would like to thank you for your constructive comments that led to the improvement of the current manuscript. To present Table 1 more informative, some definitions (Lines 131-135) based on clinicopathological data of PCa patients were added in Table 1 to be more informative.

  1. In the materials and methods section, method lack references.

We would like to thank you for your constructive comments that led to the improvement of the current manuscript. In the materials and methods section, the authors add the following references in the text:

  • Purification of RNA using TRIzol (TRI reagent), DOI: 10.1101/pdb.prot5439 (Line 323)
  • Poly(A) tag library construction from 10 ng total RNA, DOI: 10.1007/978-1-4939-2175-1_16 (Line 335)
  • Real-time quantitative PCR: A tool for absolute and relative quantification, DOI: 10.1002/bmb.21552 (Line 356)
  • Biostatistics Series Module 3: Comparing Groups: Numerical Variables, DOI: 10.4103/0019-5154.182416 (Line 370)

  1. The objectives should be better explained at the end of the introduction.

We would like to thank you for your constructive comments that led to the improvement of the current manuscript. In the introduction section, the authors add the following objectives in the text (Lines 106-116): The pursuit of more efficient and less harmful treatment options remains a focal point in contemporary oncology research. This study aims to evaluate the levels of 28S rRFs as potential prognostic indicators to distinguish between prostate cancer (PCa) and benign prostate hyperplasia (BPH). In addition to supplementing established clinicopathological factors, the researchers aim to clarify the prognostic significance of altered 28S rRF expression levels in predicting post-treatment disease relapse, potentially serving as an independent prognostic marker. In conclusion, this study seeks to contribute to the evolving field by investigating new molecular markers in prostate cancer, specifically focusing on fragments derived from 28S rRNA. These markers hold the potential to significantly transform prognosis methodologies in cancer care.”

  1. How many primers are needed for the process of reverse transcription of RNA?

We would like to thank you for your constructive comments that led to the improvement of the current manuscript. Reverse transcription of RNA usually requires the use of a single primer. The reverse transcriptase enzyme proceeds to manufacture complementary DNA (cDNA) once this primer, an oligonucleotide, hybridizes to the RNA template. This primer anneals to the RNA molecule at the specific area of interest (such as the 3' polyadenylated tail or a specific region within the RNA sequence). It is also known as a reverse transcription primer or a gene-specific primer. This primer serves as the starting point for the reverse transcriptase to synthesis the complementary DNA strand after it has annealed. For certain objectives or to capture distinct portions of the RNA molecule, researchers may occasionally use additional primers or changes to preexisting primers. However, one primer annealing to the RNA sequence is usually adequate for the fundamental step of reverse transcription, which synthesizes cDNA from RNA.

  1. What is the target length of M-MLV reverse transcriptase?

We would like to thank you for your constructive comments that led to the improvement of the current manuscript. The length of the target sequence does not determine the target length of M-MLV (Moloney Murine Leukemia Virus) reverse transcriptase, a commonly used enzyme in molecular biology for creating complementary DNA (cDNA) from RNA templates. Like other reverse transcriptases, M-MLV reverse transcriptase lacks intrinsic specificity for a certain target sequence length. Regardless of the length of the RNA molecule, it catalyzes the synthesis of a complementary DNA strand from an RNA template. In reverse transcription procedures, the enzyme is frequently employed to produce cDNA for use in PCR, cloning, gene expression analysis, and sequencing, among other downstream processes. The whole quality, integrity, and purity of the RNA template being used in the reverse transcription procedure determines the enzyme's functionality rather than the length of the target RNA sequence. Consequently, M-MLV reverse transcriptase is a flexible tool in molecular biology research since it may be used to transcribe different lengths of RNA into cDNA. Nonetheless, first-strand cDNA up to 7 kb can be synthesized using the M-MLV reverse transcriptase (Invitrogen, Thermo Fisher Scientific Inc., Waltham, MA, USA).

  1. Why is SYBR green I assay applied to a number of conditions where a tumour suppressor gene is involved in inherited susceptibility to cancer?

We would like to thank you for your constructive comments that led to the improvement of the current manuscript. The SYBR Green I assay's cost-effectiveness and versatility in measuring DNA amplification render it useful in conditions involving tumor suppressor genes associated with hereditary susceptibility to cancer. This assay is beneficial for several reasons in tumor suppressor gene investigations linked to inherited cancer risk. Firstly, SYBR Green I is a non-specific DNA-binding dye that fluoresces upon intercalation into double-stranded DNA. Researchers can examine various genes or regions of interest, including tumor suppressor genes connected to inherited cancer risk, using it for a variety of PCR applications. Furthermore, the SYBR Green I assay is cost-effective than other probe-based methods because it does not require for the creation and acquisition of probes. This makes it especially useful for research that involve several gene targets or where financial considerations are involved. Because genetic mutations are so variable, researchers examining tumor suppressor genes linked to inherited cancer risk frequently require flexibility in their assays. The target genes can be modified, deleted, or mutated in a variety of ways using SYBR Green I. The presented assay enables quantitative assessment of DNA amplification, offering insights into the presence, absence, or alterations in the tumor suppressor gene sequences. This quantitative data can be used to detect abnormal gene expression levels, identify mutations, and evaluate the expression levels of a gene. Furthermore, SYBR Green I enables a rapid initial screening of possible genetic variations or modifications in tumor suppressor genes prior to more targeted studies, which is frequent among research projects focusing on hereditary cancer susceptibility and including several gene candidates. Finally, the SYBR Green I assay's effectiveness in detecting DNA amplification provides a useful foundation for exploratory or preliminary research aimed at identifying potential genetic markers associated with inherited cancer susceptibility, enabling researchers to identify candidate genes for additional study.In summary, the SYBR Green I test is an effective choice for preliminary screenings and quantitative evaluations of tumor suppressor genes linked to hereditary cancer susceptibility due to its versatility, affordability, and capacity to quantitatively examine DNA amplification.

  1. What are the relations between threshold cycles and relative gene expression?

We would like to thank you for your constructive comments that led to the improvement of the current manuscript. In quantitative PCR (qPCR), threshold cycles (Ct values) are inversely correlated with the concentration of target nucleic acid in the sample. The PCR cycle number at which the fluorescence produced by the target's amplification exceeds a predetermined threshold is represented by the Ct value. Lower Ct values, in terms of relative gene expression, signify a larger concentration of the target nucleic acid in the sample. This may indicate that the original sample had higher levels of expression for the target gene. Higher Ct values, on the other hand, indicate a lower target nucleic acid quantity in the sample. This suggests that the gene of interest had lower expression levels in the original sample. Based on variations in Ct values, the relative expression is ascertained, offering information on the fold change or relative abundance of the target gene in relation to a reference or control sample. Prostate tumor Ct values in the study were substantially greater than those of tissue specimens from benign prostate hyperplasia (BPH). Consequently, specimens of benign prostate hyperplasia (BPH) tissues exhibited considerably higher expression of the 28S rRF gene than specimens of prostate cancers.

  1. Comments on the Quality of English Language, English spelling should be double-checked.

The manuscript was reviewed and edited by a medical reviewer proficient in English. The content of the study was extensively reviewed by the authors for grammatical and Syntax errors. Experienced scholarly writers who are native speakers have edited this manuscript and are confident about the grammar. We have proofread the manuscript again and made all the changes that were requested.

The authors wish to thank the Reviewers for their constructive comments that led to the improvement of the current manuscript.

Reviewer 4 Report

Comments and Suggestions for Authors

The paper by Diamantopoulos et al is a well-written paper about 28S rRF as a potential novel molecular marker for PCa diagnosis and prognosis. The study comprises a PCa cohort and a BPH control group. The authors claim that their results underscore the potential of 28S rRF as a novel molecular marker for PCa diagnosis and prognosis, and that further validation within larger independent cohorts are still needed to be investigated.

There are however some points that need to be considered:

2.2. Increased 28S rRF levels are correlated with unfavorable prognostic markers of PCa

-The text referring to Figure 1 only mention Fig 1 c and d. Figure 1a and b are not mentioned at all. Why?

-According to the heading 2.2, increased levels of 28rRF levels are correlated to unfavourable prognostic markers of PCa. Howcome Figure 1a, c and d show higher levels of 28S rRNA fragment-derived levels for BPH (1a), pT2 (1c) and GS 5-6 (1d), which are considered benign/less unfavourable?

-The LNCaP cell line was used for culture and RNA extraction, but there are no results displayed for the cell line. Howcome?

Comments on the Quality of English Language

The English language is fine, but unusually many words are used that are not commonly seen in scientific texts within this field.

Author Response

Reviewer #4 (Comments to the Author):

  1. In the 2.2. Increased 28S rRF levels are correlated with unfavorable prognostic markers of PCa The text referring to Figure 1 only mention Fig 1 c and d. Figure 1a and b are not mentioned at all. Why? According to the heading 2.2, increased levels of 28rRF levels are correlated to unfavourable prognostic markers of PCa. Howcome Figure 1a, c and d show higher levels of 28S rRNA fragment-derived levels for BPH (1a), pT2 (1c) and GS 5-6 (1d), which are considered benign/less unfavourable?

The authors wish to thank the Reviewer for their constructive comments that led to the improvement of the current manuscript. Tthe authors changed the title “2.2. Increased 28S rRF levels are correlated with unfavorable prognostic markers of PCa” to “2.2. 28S rRF expression levels in PCa compared to BPH patients” (Line 136). This change has as purpose to correlate the title of the heading 2.2 with Figure 1.A-D. Secondly, we would like to inform you that the authors mentioned Figure 1a and b in the manuscript as part of the paragraph with heading 2.2. (Line 139-145).

  1. The LNCaP cell line was used for culture and RNA extraction, but there are no results displayed for the cell line. Howcome?

The authors wish to thank the Reviewer for their constructive comments that led to the improvement of the current manuscript. Furthermore, as far as your second comment is concerned, it is true that while the LNCaP cell line was utilized for culture and RNA extraction, the results obtained from these experiments have not been included in the specific section or figures of the manuscript that you are currently reviewing. The LNCaP cell line experiments have been conducted to validate specific methodologies or for preliminary assessments, but the focus of the presented study was primarily on patient samples, so the results of cell line culture have not been included in the main discussion. Under this prism, the authors selected to omit certain results that seem less relevant to the central argument or narrative of this study and presented results obtained from patient samples rather than cell line experiments.

  1. Comments on the Quality of English Language. The English language is fine, but unusually many words are used that are not commonly seen in scientific texts within this field.

The content of the study was extensively reviewed by the authors for grammar and syntax errors.

The authors wish to thank the Reviewers for their constructive comments that led to the improvement of the current manuscript.
